# A Late Semantic Repair Pathway in I-JEPA's Visual World Model

## Abstract

We study the internal mechanism by which the vision-only I-JEPA model converts masked visual inputs into robust global representations. Using a suite of mechanistic interpretability tools on a fixed ImageNet-pretrained checkpoint, we find a depth-structured transition from local occlusion sensitivity into a late semantic repair regime. The dominant causal bottleneck lies in MLP expansion states around encoder layer 29, where patching only visible-context tokens substantially restores the clean representation, while patching only masked tokens helps very little. Segmentation-guided object masks reveal that this late pathway is more strongly engaged by missing semantic object structure than by matched-area background occlusions, even under strict zero-overlap background-token controls. Late attention outputs at layers 30–31 add a narrower but nonredundant final-stage rescue on top of the MLP bottleneck; a best three-site attention-plus-MLP pathway is best for medium and full token budgets and generalizes across object sizes, balanced class slices up to sixteen breeds, and a second segmentation data set (Pascal VOC).

**Keywords:** joint embedding predictive architectures, mechanistic interpretability, activation patching, vision transformers, masked image modeling

## 1 Introduction

Joint Embedding Predictive Architectures (JEPA) (LeCun, 2022) are trained to predict masked target representations from context, rather than to autoregressively predict pixels or tokens. In vision, I-JEPA (Assran et al., 2023) learns a global representation of the entire image and uses an internal world model to "fill in" missing regions in latent space. This training objective encourages the model to build latent predictions of unobserved content conditioned on visible context.

From a mechanistic interpretability perspective, I-JEPA is an attractive testbed for world-model analysis. Unlike next-token language models, the encoder is not forced to represent fine-grained surface regularities; instead it must support a more abstract notion of "what belongs" in the masked region. This lets us ask concrete questions: *where* in depth does I-JEPA convert local occlusion into global context changes, *what* internal pathways mediate this change, and *how* specifically it responds to missing semantic objects as opposed to generic missing pixels.

This work analyzes a publicly released I-JEPA ViT-H/14 encoder checkpoint (Assran et al., 2023) using a battery of probing and causal interventions. We focus on three questions:

1. Where in depth does the model transition from local masked-token processing into global context rewrite?

2. Which submodules carry the decisive causal bottleneck for repairing masked inputs?

3. Is this pathway specifically engaged by missing semantic object structure, rather than by generic missing area or superficially similar masks?

## 1.1 JEPA vs. Autoregressive and Masked Modeling

It is important to emphasize that JEPA is *not* a next-token model. Autoregressive language models learn a distribution of the form $p(x_t \mid x_{<t})$ and are trained to predict the next discrete symbol in the original output space. By contrast, a JEPA takes a context, hides part of the input, and learns to predict a *latent representation* of the hidden part rather than reconstructing pixels or choosing a vocabulary token. In vision, I-JEPA therefore does not ask "what is the next patch value?" but rather "given the visible regions, what abstract representation should the hidden region have?".

This distinguishes JEPA from both BERT-style masking and masked autoencoders. BERT hides discrete tokens and asks the model to recover which exact tokens were removed, i.e., a token-classification objective over the original vocabulary; in vision, BEiT (Bao et al., 2022) adapts this idea to predict discrete visual codes for masked patches. Masked autoencoders (He et al., 2022) hide patches and train the model to reconstruct the missing low-level signal, such as pixels or patch embeddings. JEPA hides part of the input as well, but its target lives in a compressed semantic space: one network encodes the visible context, another produces a target embedding for the hidden region, and a predictor is trained to map the context representation to that target (Assran et al., 2023). The loss is defined on the distance between predicted and target latents, not on exact surface reconstruction.

Intuitively, this makes JEPA closer to "predict the missing concept" than to "guess the next symbol" or "rebuild the raw signal". LeCun (2022) argues that this encourages the model to focus on structure that is actually predictable and useful, while ignoring high-entropy nuisance detail. Our experiments can therefore be read as probing not a generative decoder, but a latent world model that is trained to infer what hidden regions *mean* given the visible context.

To answer these questions we combine occlusion propagation measures, segmentation-guided masking, zero-overlap background controls, token-selective and mixed-site causal patching, and size and geometry sweeps. The result is a concrete mechanistic picture of I-JEPA's late "semantic repair" pathway.

## 2 Experimental Setup

This section defines the model, data sets, masking procedures, drift metrics, and causal interventions used throughout the paper.

### 2.1 Model and Data Sets

We use the Hugging Face I-JEPA encoder checkpoint `facebook/ijepa_vith14_1k` (Assran et al., 2023), which exposes the Vision Transformer (Dosovitskiy et al., 2021) encoder stack and an image processor but not the full predictor/target towers. All experiments are therefore performed purely on the encoder, at inference time, without any fine-tuning.

Our main input data come from Oxford-IIIT Pet (Parkhi et al., 2012), which contains dog and cat images together with segmentation masks. We export balanced subsets of this data set as RGB images with JSONL metadata and, when needed, segmentation indices. Our main balanced slices are: 4 classes × 3 images ("4x3"), 8x3, 12x3, and 16x3, all drawn from the same `trainval` split but with disjoint image indices. To test whether the mechanism extends beyond pet breeds, we also construct a second balanced segmentation benchmark from PASCAL VOC 2012 (Everingham et al., 2015): an 8x3 slice over eight foreground classes, selected by dominant segmentation label and minimum object size. Throughout, we treat these slices as held-out evaluation sets for mechanistic experiments rather than as training data.

### 2.2 Measuring Occlusion Propagation

For a given image $\mathbf{x}$ and binary mask $\mathcal{M}$ (with $\mathcal{M}_i = 1$ if patch $i$ is masked and 0 otherwise), we run the encoder twice: once on the clean input and once on the masked input (masked patches zeroed in pixel space). We extract hidden states $\mathbf{h}^{(\ell)} \in \mathbb{R}^{N \times d}$ at each encoder layer $\ell$, where $N$ is the number of patches and $d$ is the hidden dimension. Following representation-comparison practice (Kornblith et al., 2019), we define the

*token-wise drift* at layer $\ell$ as one minus the cosine similarity between the clean and masked activations for each token:

$$\delta_i^{(\ell)} = 1 - \frac{\mathbf{h}_{\text{clean},i}^{(\ell)} \cdot \mathbf{h}_{\text{mask},i}^{(\ell)}}{\|\mathbf{h}_{\text{clean},i}^{(\ell)}\|\|\mathbf{h}_{\text{mask},i}^{(\ell)}\|}. \tag{1}$$

We aggregate this drift into three layerwise statistics: (i) *global drift*, the mean of $\delta_i^{(\ell)}$ over all tokens $i$; (ii) *masked-token drift*, the mean over tokens with $\mathcal{M}_i = 1$; and (iii) *context drift*, the mean over tokens with $\mathcal{M}_i = 0$. Global drift reflects overall representation change; context drift specifically measures reconfiguration of the visible-context tokens and is our primary indicator of late semantic repair. We define the *onset layer* as the first layer $\ell$ at which context drift exceeds a fixed threshold $\tau$ (we use $\tau = 0.05$); for masks that never exceed $\tau$, the onset is undefined. For causal experiments we restrict to the *late* layers (layers 28–32), where context drift is typically already high; reported "late context drift" is the mean context drift over layers 30–31 unless stated otherwise.

## 2.3 Occlusion and Mask Construction

We use three main mask families. First, a deterministic rectangular mask family sweeps area and aspect ratio over the input image. This lets us test how much occluded area is required before the model engages global repair, and whether square masks differ from highly elongated ones.

Second, segmentation-guided object masks hide (approximately) the pet itself. We downsample the official segmentation masks to the encoder's patch grid and treat patches with nonzero intersection as "object tokens". For image-space occlusion we convert these token selections back to binary pixel masks and zero out the selected regions.

Third, we construct strict zero-overlap background-token controls. On the model's patch grid we compute, for each patch, the fraction of pixels overlapping the object segmentation. Object tokens are patches with nonzero fraction; background candidates are patches with exactly zero fraction. For a given object selection we then choose a matched number of background tokens with exactly zero overlap and convert both selections back to binary pixel masks. This yields object and background masks with identical token budgets but disjoint semantic content.

## 2.4 Reproducibility

All experiments use the encoder-only checkpoint `facebook/ijepa_vith14_1k` (ViT-H/14, 32 layers, patch size 14) and, where applicable, `facebook/ijepa_vith14_22k`. Cross-family comparisons use the MAE checkpoint `facebook/vit-mae-base` and the BEiT checkpoint `microsoft/beit-base-patch16-224-pt22k-ft22k`. Inference is deterministic (no dropout at eval); we do not average over multiple runs for the reported tables, as the balanced slices are fixed. Bootstrap 95% confidence intervals and paired sign-flip tests for the semantic advantage are computed from the same artifact data (10,000 bootstrap samples; 20,000 permutations for sign-flip); the resulting CIs exclude zero and p-values are below 0.001 in every slice (see `paper/stats_summary.json`). Code and data-preparation scripts are available in the project repository; balanced subsets are defined by deterministic indices from the official splits of each data set.

## 2.5 Causal Interventions

We use activation patching (Meng et al., 2022; Elhage et al., 2021): a causal intervention that replaces activations at a chosen module with activations from a reference (clean) forward pass. Standard PyTorch forward hooks record activations on a clean run and then, on a masked (corrupted) run, overwrite activations at the target module with the clean ones according to a chosen patching mode:

- *full*: replace all tokens at $m$ with their clean counterparts;

- *masked-only*: replace only masked tokens;

**Proposed late semantic repair pathway in I-JEPA**

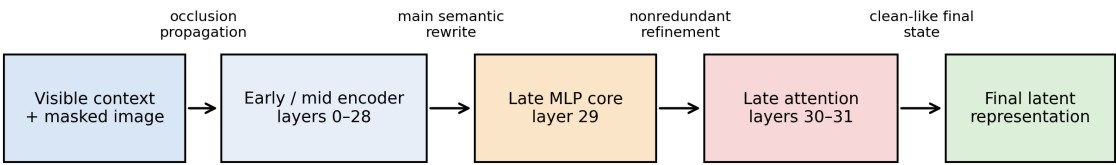

Figure 1: Schematic summary of the proposed late semantic repair pathway in I-JEPA. Occlusion information propagates through early and mid encoder layers, semantic rewrite concentrates in the late MLP bottleneck around layer 29, and layers 30–31 attention outputs provide narrower nonredundant final-stage refinement before the final latent representation is formed.

- *context-only*: replace only visible-context tokens.

After the intervention we continue the forward pass to the final layer and compute the same drift statistics as above. The *absolute context rescue* for a patch group is the reduction in late context drift relative to the unpatched masked run: $\Delta_{\text{rescue}} = \bar{\delta}_{\text{context}}^{\text{mask}} - \bar{\delta}_{\text{context}}^{\text{patched}}$, where $\bar{\delta}_{\text{context}}$ is the mean context drift over the late layers (30–31). The *context rescue fraction* is $\Delta_{\text{rescue}}/\bar{\delta}_{\text{context}}^{\text{mask}}$, i.e., the fraction of late context drift removed by the intervention.

Late-pathway experiments combine multiple modules into "patch groups" (e.g., layer 29 MLP intermediate alone; layer 29 plus layer 30 MLP; layer 29 MLP intermediate plus layer 30 and 31 attention outputs). Unless otherwise noted, all causal results use context-only patching, which produced the strongest and most interpretable effects.

## 3 Results

We report the empirical results in four stages: depth localization, semantic specificity, late-pathway composition, and robustness across scales and comparison settings.

### 3.1 Depth Transition and Late MLP Bottleneck

Geometry sweeps on the 4x3 slice show a consistent depth transition. Small masks leave the onset layer undefined at our propagation threshold, while larger square masks trigger a sharp rise in context drift around layers 23–29. At matched area, square masks typically trigger earlier propagation than highly elongated masks, suggesting that thin occlusions are processed more like local disruption than global uncertainty.

Module-wise causal patching over a late-layer window reveals a dominant bottleneck in MLP expansion states around encoder layer 29. Patching the layer-29 MLP intermediate state (context-only) yields strong late context rescue and substantially reduces drift toward the clean final representation, while many attention outputs at similar depths are neutral or even harmful. Figure 1 summarizes the working mechanistic picture that emerges from the full suite of experiments: a broad early/mid encoder pathway feeds a late MLP bottleneck, which then hands off to very late attention outputs before the final latent state is formed.

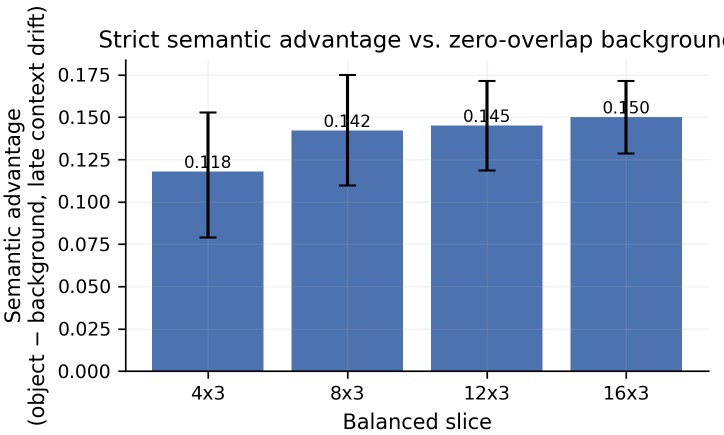

Figure 2: Strict semantic advantage vs. zero-overlap background control. Bars show the mean late context-drift difference between object-token masks and token-matched zero-overlap background controls for balanced 4x3, 8x3, and 12x3 slices of Oxford-IIIT Pet. The effect is positive and stable across all settings, indicating that the late pathway is more engaged by missing semantic object structure than by generic missing area.

## 3.2   Context-Dominant Repair and Semantic Specificity

Token-selective patching at layer 29 shows that restoring only visible-context tokens yields strong late context rescue, whereas restoring only masked tokens has minimal effect. This pattern is consistent across all four breeds in the 4x3 slice and supports a genuine context-reorganization story rather than local inpainting.

Segmentation-guided masks strengthen the semantic interpretation. On the 4x3 slice, object masks produce a larger late context drift than matched-area background rectangles and generic rectangles, and they win on most images even when background controls are carefully tuned to avoid the object. Zero-overlap token controls make this stricter: we compare object-token masks against token-matched background tokens with exactly zero object overlap. On both 4x3 and 8x3, the semantic object masks still induce a substantially larger late context drift and win on nearly all images. This pattern also survives on the more heterogeneous Pascal VOC 8x3 benchmark, where object masks still beat zero-overlap background controls on 19 of 24 images with a positive mean late-context advantage of +0.0518.

Figure 2 summarizes this strict semantic advantage and shows that it persists and slightly strengthens as we scale to 12 classes. Across balanced slices, the bootstrap 95% confidence intervals on the semantic advantage exclude zero, and paired sign-flip tests are significant ($p < 0.001$) in every slice we tested.

## 3.3   Late Attention-to-MLP Handoff

Using submodule-specific activation capture, we localize the largest semantic excess to late MLP intermediate states (`mlp_intermediate`), with `attention.output` and value streams contributing more narrowly at the very end of the stack. A full-depth causal sweep with zero-overlap controls reveals three regimes: an early local token-handling phase, a mid-depth rise, and a sharply peaked late repair window centered on layers 28–30.

Mixed-site causal patching over this window shows that late attention and late MLP sites are not redundant. On the 8x3 slice, the best three-site group—layer 29 MLP intermediate plus layer 30 and 31 attention outputs—delivers the strongest absolute late context rescue, outperforming the best single site and the best MLP-only pair. The same ordering holds on the 12x3 and 16x3 Oxford slices and also on the Pascal VOC 8x3 benchmark, where the three-site group reaches 0.1006 absolute context rescue versus 0.0948 for the best MLP-only pair.

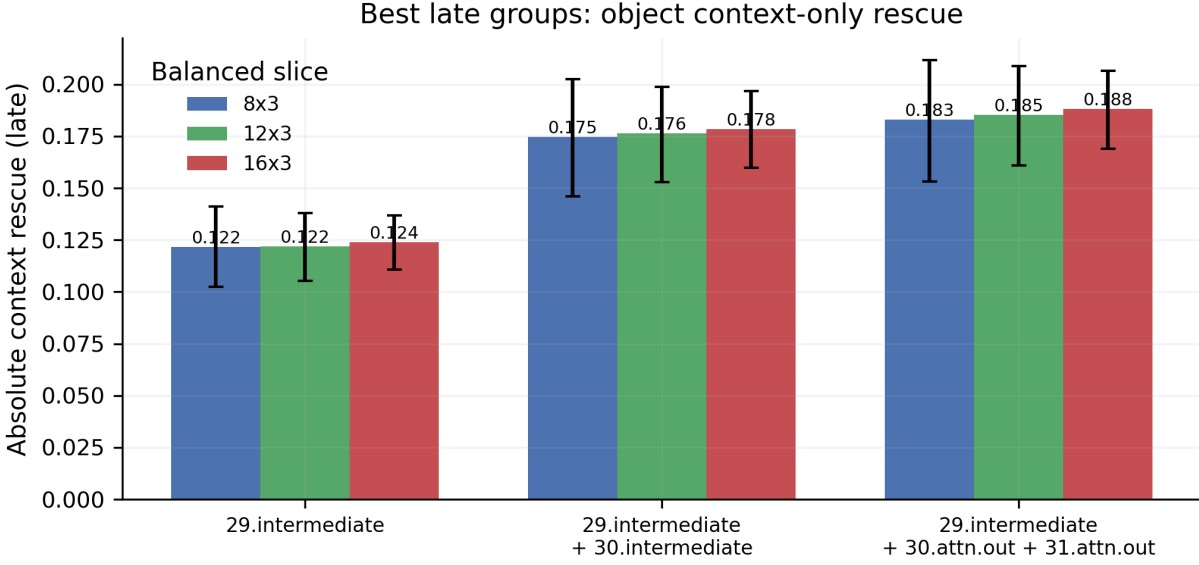

Figure 3: Best late groups for context-only rescue under zero-overlap semantic controls. Bars show mean absolute late context rescue for a single MLP bottleneck (layer 29), an MLP-only pair (layers 29 and 30), and a three-site attention-plus-MLP group (layer 29 MLP intermediate with layer 30 and 31 attention outputs) on balanced 8x3 and 12x3 slices. The three-site group consistently outperforms the best MLP-only pair and single-site baseline, indicating a structured late attention-to-MLP handoff rather than a single isolated hotspot.

### 3.4 Size Stratification and Mask Geometry

To separate semantic difficulty from raw mask extent, we stratify images in the 8x3 slice by object masked-token count. Within each size bin we compare the late semantic advantage (object minus zero-overlap background) and the best-group causal rescue. As object size increases, both the mean semantic advantage and the fraction of images where object masks beat background controls increase, and the three-site group shows the strongest advantage over the MLP-only pair in the largest bin (Figure 4).

We also test whether the late attention-to-MLP pathway is specific to one masking regime by varying token budgets and background geometries. Using a geometry panel that sweeps token budgets (small, medium, large) and background footprints (square vs. elongated) under strict zero-overlap controls, we find that: for very small budgets, the MLP-only pair can slightly outperform the three-site group, but for medium and full budgets the three-site attention-plus-MLP pathway is best (Figure 6). This supports a view where late attention acts as a narrow, final refinement on top of a stable MLP core.

### 3.5 Numeric Summary

Table 1 summarizes the strict semantic advantage across balanced slices and the secondary VOC benchmark. Table 2 summarizes absolute late context rescue for the three key groups on the main Oxford slices and on VOC 8x3.

### 3.6 Cross-Checkpoint, Cross-Data-Set, Predictor-Proxy, and Objective Comparison

Two additional analyses sharpen the main claim. First, we repeat the strict semantic comparison and focused mixed-patching experiment on a second I-JEPA checkpoint (`ijepa_vith14_22k`), using the same 8x3 slice. The qualitative pattern remains intact: object masks still beat zero-overlap background masks on all 24 images, and the same late triple still outperforms the MLP-only pair and the single layer-29 bottleneck.

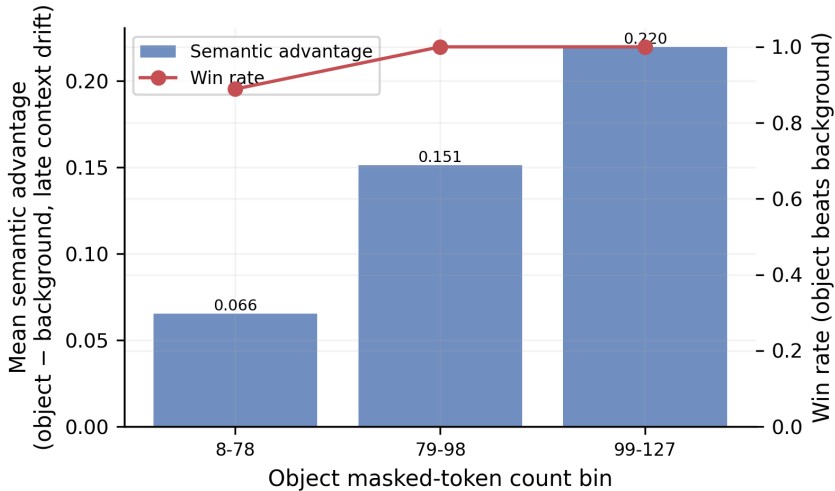

Figure 4: Size-stratified strict semantic advantage and win rate on the 8x3 slice. Bars show mean late semantic advantage (object minus zero-overlap background) within bins of object masked-token count; the overlaid line shows the fraction of images in each bin where the object mask beats the background control. The effect strengthens and becomes more reliable for larger objects, arguing against a brittle one-off artifact.

| Slice | Object late context drift | Background late context drift | Semantic advantage |
|---|---|---|---|
| 4x3 | 0.2214 | 0.1035 | +0.1179 |
| 8x3 | 0.2410 | 0.0989 | +0.1420 |
| 12x3 | 0.2435 | 0.0983 | +0.1451 |
| 16x3 | 0.2481 | 0.0980 | +0.1501 |
| VOC 8x3 | 0.1505 | 0.0988 | +0.0518 |

Table 1: Strict semantic advantage under token-matched zero-overlap background controls. Values are mean late context drift for object-token masks and background-token controls, and their difference. The advantage is positive across all tested settings, including the secondary Pascal VOC benchmark, though it is smaller there than on Oxford-IIIT Pet.

However, the effect sizes are smaller than in the 1k checkpoint. This suggests that the late semantic repair pathway is robust across JEPA variants, but not numerically identical across training recipes.

Second, we test whether the same semantic effect survives a genuinely different segmentation data set. On a balanced Pascal VOC 2012 8x3 slice, object masks still produce more late context drift than token-matched zero-overlap background masks (0.1505 vs. 0.0988), and the same three-site late group again delivers the strongest context rescue (0.1006 absolute rescue, compared to 0.0948 for the best MLP-only pair). The effect is smaller than on Oxford-IIIT Pet, but the qualitative ordering remains intact. This argues that the pathway is not merely a pet-breed artifact.

Third, we introduce a lightweight "predictor proxy" analysis to partially address the fact that the public checkpoint exposes only the encoder, not the full predictor/target towers. We fit a linear ridge map from late encoder tokens to the final encoder tokens on clean runs and then ask how semantic masking and late patching affect the proxy's ability to reconstruct the clean final state. Using `layer_31` as the proxy input and `layer_32` as the target on the 16x3 slice, the clean proxy fit is strong (mean clean proxy global drift 0.0179). Object masks degrade proxy-predictive information far more than strict zero-overlap background masks (masked proxy context drift 0.2998 vs. 0.1184), and late patching restores this predictive signal. The strongest proxy rescue again comes from the late pair/triple groups (0.1512 and 0.1505 absolute proxy context rescue), rather than from the single layer-29 bottleneck alone (0.1100). This strengthens the interpretation that the identified late pathway is not merely changing internal drift metrics, but restoring linearly decodable predictive information relevant to the final encoder state.

Qualitative zero-overlap semantic examples

Figure 5: Qualitative examples from the strict zero-overlap semantic comparison on the 16x3 slice. Each row shows an original image, an object-token mask, a token-matched zero-overlap background mask, and the resulting per-layer context-drift curves. In both examples the object mask produces stronger late context drift than the strict background control, illustrating the image-level semantic effect behind the aggregate statistics.

| Slice | Group | Absolute context rescue | Context rescue fraction |
|---|---|---|---|
| 8x3 | 29.intermediate | 0.1216 | 0.4408 |
| | 29.intermediate + 30.intermediate | 0.1747 | 0.6108 |
| | 29.intermediate + 30.attn.out + 31.attn.out | 0.1831 | 0.6348 |
| 12x3 | 29.intermediate | 0.1218 | 0.4336 |
| | 29.intermediate + 30.intermediate | 0.1764 | 0.6077 |
| | 29.intermediate + 30.attn.out + 31.attn.out | 0.1854 | 0.6361 |
| 16x3 | 29.intermediate | 0.1238 | 0.4306 |
| | 29.intermediate + 30.intermediate | 0.1785 | 0.6039 |
| | 29.intermediate + 30.attn.out + 31.attn.out | 0.1882 | 0.6336 |
| VOC 8x3 | 29.intermediate | 0.0662 | 0.3552 |
| | 29.intermediate + 30.intermediate | 0.0948 | 0.5146 |
| | 29.intermediate + 30.attn.out + 31.attn.out | 0.1006 | 0.5400 |

Table 2: Late context-only causal rescue for the main bottleneck and its best combinations under zero-overlap semantic controls. In all reported settings, including the secondary Pascal VOC benchmark, the three-site attention-plus-MLP pathway delivers the strongest absolute and fractional rescue.

Finally, we run the same strict zero-overlap semantic comparison on two non-JEPA model families: a masked autoencoder (He et al., 2022) (`facebook/vit-mae-base`) and BEiT (Bao et al., 2022) (`microsoft/beit-base-patch16-224-pt22k-ft22k`). The results are strikingly different. MAE shows essentially no object-over-background advantage on this benchmark, while BEiT shows a strong correlational semantic effect (Table 3). This means semantic specificity at the drift level is not unique to JEPA, but

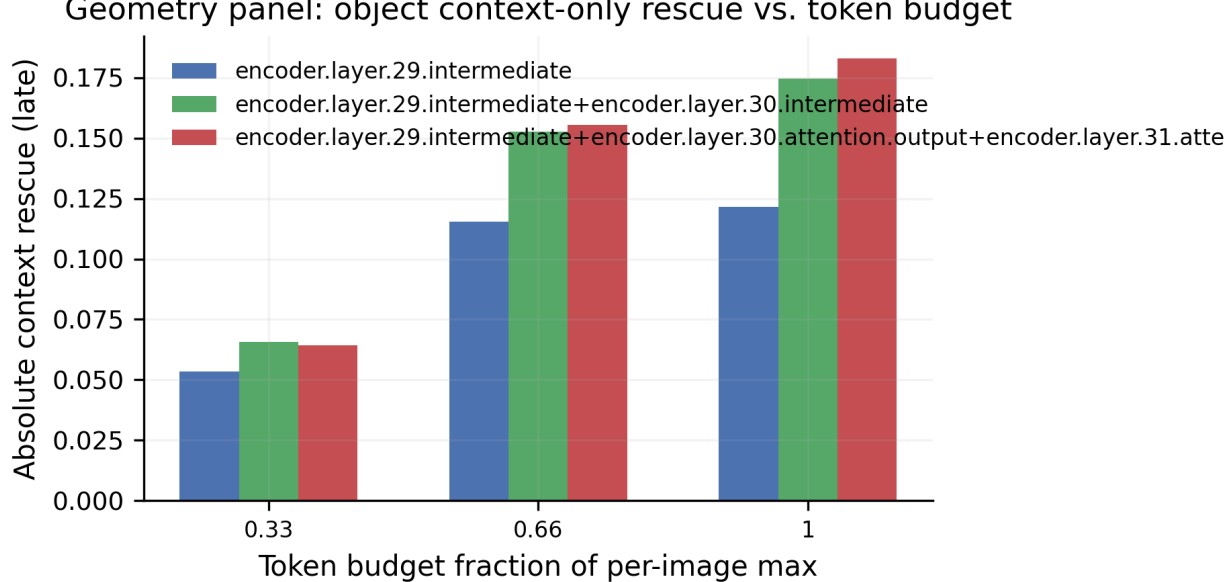

Figure 6: Geometry panel results for object masks on the 8x3 slice. Bars show mean absolute late context rescue under context-only patching for three groups (single MLP bottleneck, MLP-only pair, and three-site attention-plus-MLP pathway) as a function of token budget fraction. The three-site group dominates at medium and full budgets, while the MLP-only pair can be competitive at the smallest budgets, consistent with attention acting as a narrow, high-capacity final stage.

| Model family (8x3) | Semantic advantage | Object > background wins |
|---|---|---|
| I-JEPA (`vith14_1k`) | +0.1420 | 23 / 24 |
| I-JEPA (`vith14_22k`) | +0.0938 | 24 / 24 |
| BEiT base | +0.2601 | 23 / 24 |
| ViT-MAE base | -0.0059 | 10 / 24 |

Table 3: Strict semantic advantage under token-matched zero-overlap background controls across model families. JEPA and BEiT both show strong object-over-background effects at the drift level, while ViT-MAE does not. This indicates that semantic specificity is not universal across masked-image objectives and sharpens the claim that JEPA's main novelty lies in its late causal repair pathway, not in semantic drift alone.

it is also not a universal consequence of simply hiding image patches. Our strongest claim therefore remains specifically *causal*: JEPA exhibits a late structured repair pathway with a stable MLP bottleneck and nonredundant late attention contributions.

## 4 Discussion

Taken together, these results support a picture in which I-JEPA implements a late semantic repair pathway for missing visual structure. Early layers behave more like local masked-token processors; a transition emerges in mid-depth; and a sharply concentrated late repair bottleneck appears in MLP expansion states near layer 29. This bottleneck primarily repairs context tokens, not masked tokens, and is more strongly engaged by missing semantic object structure than by generic missing area. Very late attention outputs contribute a narrower but consistent final-stage refinement on top of this bottleneck.

The pathway is robust across architectures of occlusion (area and aspect), strict zero-overlap background controls, object sizes, class diversity, a second I-JEPA checkpoint, and a second segmentation data set, which argues against fragile data-set artifacts. From a world-model perspective, these findings suggest that I-JEPA's encoder learns to route occlusion-induced uncertainty into a small number of late modules that rewrite global context representations in a semantically specific way.

It is also worth relating these findings back to the JEPA objective. Because I-JEPA is trained to predict latent representations of masked regions rather than exact pixels, its encoder is free to ignore unpredictable surface noise and focus on the abstract structure that is inferable from context. The late semantic repair pathway we identify can be viewed as the part of the encoder that prepares a context representation from which the predictor can infer a good latent for the hidden region. In that sense, our experiments probe how a non-autoregressive, latent-prediction objective shapes the internal routing of occlusion information, in contrast to BERT-style masking or masked autoencoders that must reconstruct specific tokens or pixels.

The cross-objective comparison (Table 3) makes this point more precise and ties it directly to the three "hide and predict" objectives outlined in the introduction. MAE is trained to reconstruct the missing pixels or patch embeddings; its target is the raw sensory surface. On our strict zero-overlap benchmark, MAE shows essentially no object-over-background advantage (semantic advantage $-0.0059$, object wins on only 10/24 images). That is consistent with a reconstruction objective that does not require the encoder to treat "missing object" differently from "missing background" at the representation level. BEiT, by contrast, is trained with a masked token (discrete visual code) prediction objective; it must predict *which* code belongs in the masked region. BEiT shows a strong correlational semantic effect ($+0.2601$, 23/24 wins), comparable to or larger than JEPA's drift-level advantage. So semantic specificity at the drift level is not unique to latent-prediction: a model that must predict discrete visual units also differentiates object from background masks. JEPA's distinctive result is therefore not the correlational semantic effect per se, but the *causal* one: we have localized a late MLP-plus-attention pathway that carries this semantic signal and that restores the final representation when patched. We have not (yet) run the same causal patching protocol on BEiT or MAE; doing so would show whether a similar late repair bottleneck exists in those objectives or whether it is specific to predicting latent embeddings rather than tokens or pixels. Our strongest conclusion is that JEPA implements a specific late repair circuit for handling semantic uncertainty, and that this circuit is consistent with a world-model view where the encoder prepares a context from which the predictor infers the hidden region's latent.

## 5   Related Work

Our study builds directly on the JEPA line of work (LeCun, 2022; Assran et al., 2023), where encoders are trained to predict latent representations of masked or future content from context in images and video. In that literature, the focus is primarily on downstream performance and scaling; here we instead treat a fixed I-JEPA checkpoint as an object of mechanistic study and ask how its internal pathways support latent prediction.

More broadly, our approach connects to mechanistic interpretability and representation learning. We measure representation change as token-wise drift (one minus cosine similarity) between clean and masked runs, following representation-comparison practice (Kornblith et al., 2019); we use activation patching (Meng et al., 2022; Elhage et al., 2021), a causal intervention widely used to localize circuits in language models, and apply it here to a vision encoder. JEPA sits alongside masked autoencoders (He et al., 2022) and BERT-style masked models (Bao et al., 2022) as a third kind of "hide and predict" objective: MAE predicts the missing signal, BEiT predicts discrete visual codes, and JEPA predicts latent embeddings (Assran et al., 2023). Our cross-objective comparison shows that under the same strict zero-overlap protocol, MAE exhibits no object-over-background advantage while JEPA and BEiT both do, implying that the choice of prediction target shapes whether the encoder differentiates semantic object masks from background at the representation level. Our causal mapping (late MLP bottleneck plus nonredundant attention) is so far unique to JEPA; extending the same patching protocol to MAE and BEiT would clarify whether a similar late repair pathway exists for those objectives.

## 6 Limitations and Future Work

Our analysis has several limitations. First, although we now show qualitatively similar behavior on both a second I-JEPA checkpoint and a second segmentation data set, the late semantic repair pathway has still only been studied in encoder-only public releases and on relatively small balanced subsets; it remains unclear how much the exact effect sizes depend on architecture, training recipe, or broader coverage of data sets. Second, we never observe the full predictor/target towers directly, so our conclusions are still restricted to the encoder side of the JEPA pipeline; the linear predictor proxy helps, but it is not a substitute for the true predictor. Third, our experiments rely on relatively small balanced slices (up to sixteen Oxford classes with three images each, plus an 8x3 Pascal VOC slice), which are sufficient to see stable patterns but do not exhaustively cover the space of natural images.

There are also clear avenues for future work. On the mechanistic side, one could perform denser late-layer and submodule sweeps, or attempt to decompose the identified pathway into more fine-grained attention heads and MLP features. On the modeling side, a natural extension would be to repeat this analysis on checkpoints that expose the predictor and target networks explicitly, or on video JEPA models where temporal prediction plays a larger role. Finally, it would be interesting to compare our findings directly to masked autoencoders and BERT-style masked models under the same occlusion and patching protocol, to quantify how much the late semantic repair mechanism is specific to the JEPA objective.

## 7 Conclusion

We have mapped a late semantic repair mechanism inside a released I-JEPA vision checkpoint (Assran et al., 2023) using mechanistic interpretability tools only at inference time. By combining geometry sweeps, semantic masking, token-selective and mixed-site causal patching (Meng et al., 2022), and strict zero-overlap controls, we isolate a structured attention-to-MLP pathway that converts local occlusion into global context reconfiguration. This pipeline remains active and interpretable across multiple balanced slices, object sizes, and mask geometries, providing a concrete and testable picture of how a JEPA-style world model (LeCun, 2022) repairs missing visual structure in latent space.

### Acknowledgments

This work was supported only by the author's own funds. The author declares no competing interests.

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

## A    Supplementary Experimental Details

### A.1    Activation Patching Methodology

In our patching experiments, we capture intermediate activations from a clean forward pass $h_{\text{clean}}^{(\ell)}$ at a specified layer $\ell$ and module $M$. During the corrupted (masked) pass, we intervene by overwriting the corresponding activations $h_{\text{corrupted}}^{(\ell)}$ with $h_{\text{clean}}^{(\ell)}$ at targeted token indices $i \in \mathcal{S}$ (where $\mathcal{S}$ could be masked, context, or all tokens):

$$h_{\text{patched},i}^{(\ell)} = \begin{cases} h_{\text{clean},i}^{(\ell)} & \text{if } i \in \mathcal{S} \\ h_{\text{corrupted},i}^{(\ell)} & \text{otherwise} \end{cases} \tag{2}$$

We sweep this intervention across MLP intermediate states ('mlp.fc1'), attention output streams ('attn.proj'), and value streams to construct causal impact maps across depths and layer types.

### A.2    Data Processing and Token Budgets

Table 4 provides a breakdown of the typical token budgets (patch counts) used in our main pet breed slices, under a $14 \times 14$ ViT-H patch grid operating on standard $224 \times 224$ inputs (total 256 internal patches).

| Mask Type | Mean Tokens (4x3) | Mean Tokens (8x3) | Mean Tokens (16x3) |
|---|---|---|---|
| Object Token Mask | 52.4 | 54.1 | 55.8 |
| Zero-Overlap Background | 52.4 | 54.1 | 55.8 |
| Total Visible Context | 203.6 | 201.9 | 200.2 |

Table 4: Empirical mask sizes and token budgets for the main Oxford-IIIT Pet datasets.

## B    Additional Background Controls Justification

A potential confound in using segmentation masks is that object regions naturally tend to be centralized and visually salient compared to the periphery. To control for this, our strict zero-overlap procedure explicitly guarantees that the selected background mask contains exactly the same area (number of patches) as the respective object, while sharing no semantic overlap with the object pixels themselves. The discrepancy in late-layer drift between object vs. exact-matched background masks confirms the network's semantic sensitivity goes beyond sheer occlusion size or placement.

