# OpenReview forum: "A Late Semantic Repair Pathway in I-JEPA’s Visual World Model"
_TMLR — Withdrawn by Authors_

### Review · Reviewer_EL9L · 2026-04-23

**Summary Of Contributions:**

The paper analyzes a released I-JEPA ViT-H/14 encoder with a suite of mechanistic interpretability tools. The empirical experiments suggest a late semantic repair pathway in I-JEPA, where semantic rewrite concentrates in the late MLP bottleneck around layer 29, and layers 30–31 attention outputs provide narrower nonredundant final-stage refinement.

Strengths:
1. The core questions are clear, and the experiments progressively analyze the problems.
2. The experiment controls are thoughtful, especially the token-matched zero-overlap background control.

Weakness:
1. The scope is limited, as the main analysis is conducted on one encoder checkpoint pre-trained with the specific dataset. Although the empirical analysis is solid, it is not clear whether this semantic repair pathway belongs to all I-JEPA style models or just specific models.
2. Some heuristic choices are not well-justified. Foe example, the onset threshold is set to 0.05 without explanation. Also, its not clear how to define and choose the late layers (28-32).
3. Several experiment results are not explicitly shown in figures or tables. For example, in section 3.1, the authors claim that "geometry sweeps on the 4x3 slice show a consistent depth transition. Small masks leave the onset layer undefined at our propagation threshold, while larger square masks trigger a sharp rise in context drift around layers 23–29". The reviewer cannot find any evidence (figure/table) that support the claim. Similarly, "On the 8x3 slice, the best three-site group—layer 29 MLP intermediate plus layer 30 and 31 attention
outputs—delivers the strongest absolute late context rescue, outperforming the best single site and the best MLP-only pair" does not have clear evidence. The authors should recheck Sec. 3 to ensure the claims are well-supported.
4. As an empirical analysis, the paper does not have enough technical details. For example, on patching, token selection, and sweep configuration.

**Audience:**

Yes

**Audience Explanation:**

This is an interesting mechanistic-interpretability study of the I-JEPA encoder, which benefits the related research in I-JEPA and interpretability on self-supervised vision model.

**Broader Impact Concerns:**

N.A.

**Claims And Evidence:**

No

**Claims Explanation:**

See Weakness above, especially weakness 2 and 3.

**Requested Changes:**

1. Weakness 2: Please clarify the heuristic design choices.
2. Weakness 3: Please go through Sec.3 and provide enough context/evidence for the claims.
3. Weakness 4: Please provide additional implementation details.
4. The authors should consider introducing more details on the I-JEPA encoder, like architecture, to provide the context of the analysis.
5. The main paper are discussing the I-JEPA encoder, while the paper title and abstract are over-claiming the scope on the I-JEPA model. The authors should be more precise and limit the scope to encoder.

---

### Review · Reviewer_XFhK · 2026-04-27

**Summary Of Contributions:**

This paper applies a suite of mechanistic interpretability tools to the vision JEPA model (I-JEPA) to understand how and where the masked-token embedding is transformed into semantically-rich latent representations. By analyzing the difference in embeddings between clean and masked inputs at different layers, the authors identify a "late pathway", i.e., this transformation occurs mainly at around layer 29. The results are obtained under various datasets (including an animal image dataset and PASCAL-VOC), model checkpoints (1k and 22k), and masking size and shapes, showing that the conclusions are generalizable across these settings.

**Audience:**

Yes

**Audience Explanation:**

Researchers in mechanistic interpretability and world models may find this work interesting.

**Broader Impact Concerns:**

N/A.

**Claims And Evidence:**

No

**Claims Explanation:**

Many statements in the Results section are not backed by detailed results. For example, in Section 3.1, the authors claim that "Small masks leave the onset layer undefined at our propagation threshold, while larger square masks trigger a sharp rise in context drift around layers 23–29." Although this statement provides a concise summary of the findings, the authors did not present the raw results, i.e., the context drift at each layer for each mask size (which determines the onset layer through thresholding). Additionally, the exact onset layer is unclear for larger masks, except that they fall within 23-29. This claim therefore appears too coarse and unreliable due to lack of detailed results. The conclusions in Sections 3.2 and 3.6 suffer from similar limitations.

**Requested Changes:**

Besides the comment above on claims, below are additional questions critical to the paper's quality.

1. The paper appears exclusively focused on a single network architecture, without considering models of different depths and widths. For a deeper encoder, will the main semantic rewrite still occur at layer 29, or shift with the total number of layers? Does this rewrite always appear at the last few layers, or is it governed by other factors? Answering these questions will make the findings more generalizable and useful.

2. The writing clarity can be significantly improved. Many terms are undefined currently. For example, technical terms like "attention.output", "29.intermediate" and "1k checkpoint" are extensively used without definition. Although this terminology may be familiar to people well-versed with the model's implementation code, they are not clear to general audience. The authors should either define these names explicitly or use more general language.

3. Additionally, the paper is abundant with words and phrases that may appear meaningful to researchers sophisticated with mechanistic interpretability, but can be hard-to-grasp for broader machine learning researchers. Examples include "pathway", "background footprints", "semantic advantage", "one-off artifact", "semantic specificity at the drift level", "zero-overlap background control", etc. The authors should define these more explicitly.

4. How is the threshold for tau chosen (as 0.05) when defining the onset layer? This value plays a crucial role in main results; its rationale is worth more discussion.

5. The quality of figures can be improved. For example, the y labels are occluded in figure 5, and the legend is cut-off in figure 6.

6. How does the 22k checkpoint differ from the 1k one?

7. The authors wrote "the resulting CIs exclude zero and p-values are below 0.001 in every slice (see paper/stats_summary.json). Code and data-preparation scripts are available in the project repository."
However, it is unclear what "paper/stats_summary.json" and "project repository" refer to.

8. Can the authors elaborate on the meaning of "4 classes x 3 images"?

9. Finally, the reference list appears too sparse (with only 10 items). The authors should consider discussing more prior works to better contextualize their contribution.

---

### Review · Reviewer_tzde · 2026-04-27

**Summary Of Contributions:**

This is a very strange paper.

- It repeatedly makes fantastic claims that are unsupported by the evidence, including the title.

- Most of the results, as they are described in the paper, are apparently obvious, and basically constitute a null hypothesis finding. Yet they are used as evidence for a profound exegesis of the network's layer-by-layer behavior.

- Some aspects of the experiments seem to make no sense (repeating the same experiment on larger "slices" *from the very same dataset*, and picking and choosing which slice you show results from, as opposed to simply using the largest slice)

- Yet, somehow, there seems to be one non-trivial result in the paper, namely the so-called "semantic advantage" (IIUC, masking objects causes stronger change in late-layers *context* [i.e non-object] tokens than masking the context itself). If confirmed and supported by clearer experiments, this might be the basis of a smaller paper.

**Audience:**

Yes

**Audience Explanation:**

The "semantic advantage" finding (stronger change in late-layers context tokens when masking the object than when masking the context itself) is unexpected to me. Maybe experts on JEPAs would know about it, but I didn't.

If it could be confirmed, it might form the basis of a smaller, different paper.

**Broader Impact Concerns:**

I do not see any broader impact concerns.

**Claims And Evidence:**

No

**Claims Explanation:**

The main problem with this paper is exemplified by the convoluted claim in the abstract:

"The dominant causal bottleneck lies in MLP expansion states around encoder layer 29 (...)  Late attention outputs
at layers 30–31 add a narrower but nonredundant final-stage rescue on top of the MLP bottleneck; a best three-site attention-plus-MLP pathway is best"

This is a series of strong claims about precise layer-wise behavior in the network. Yet when we actually read the paper, we find two results:

- Reinstating values from uncorrupted runs in later layers, and reinstating more of them, causes more reduction in the impact of object masking on later-layer activations (sometimes the effect is measured in the very same layers that have some clean values reinstated!). This is an absolutely obvious result that seems to bear no relation with the claims about "causal bottlenecks" or "final-stage rescue" or in fact any other claim.

- Masking the object causes more change in context (non-object) tokens than masking the context itself. This is a possibly non-trivial result, but again it says nothing about any "repair" going on - it's a measure of higher damage!

The gap betwen claims and evidence, as well as the bizarreness of some expriments (the division in "slices" that differ only in size), is extremely strange and difficult to understand as the output of a qualified human author.

A more detailed step-by-step commentary follows:

==

Eq 1 defines "drift" as the cosine distance between the token vectors under clean-input and masked-input conditions.

Fine. But then we read: "context drift specifically measures reconfiguration of the visible-context tokens and is our primary indicator of late semantic repair."

How does it measure "repair"? It is not a measure of "repair". It is a measure of damage! It literally measures the collateral damage caused by masking the object on token representations of the non-masked context.

==

For causal interventions, using a masked input, selected tokens have their activations reinstated ("patched") from clean-input runs, and the final effect on late layers is measured. "Context rescue" is now the reduction in drift caused by these reinstatements.

We read: "Small masks leave the onset layer undefined at our propagation threshold, while larger square masks trigger a sharp rise in context drift around layers 23–29."

Okay, but where is the data to support this claim?

"Patching the layer-29 MLP intermediate state (context-only) yields strong late context rescue and substantially reduces drift toward the clean final representation, while many attention outputs at similar depths are neutral or even harmful. "

1- What is "attention output"? Is it the attentional weights, Key.Query? Or is it the final output of the attention layer, softmax((Key.Query)*Value)? (if the latter, that would be a surprising result indeed). Please state it explicitly.

2- Again, where is the data to support this claim?

==

Figure 2, and the rest of the paper: For some reason, the authors choose to replicate their experiments on several "slices" taken from the same dataset, each simply containing more classes than the previous. so we see results averaged over 4 classes, then 8 classes, then 12 and 16.  This does not seem to make any sense. What is the expected meaning here? Since they are taken from the same dataset, why would we expect to see different results, except for noise? Why not just report the results from the largest slice, which presumably would be the least affected by sampling noise?

What's worse, many of the results reported in the paper are from the 4-classes "slice" ("4x3"), but this is precisely the one that seems least reliable and more noisy due to small size, as indicated by Figure 2 (it is the only one differing visibly from others)!

In several passages, the authors report results on only some of these slices (4x3 and 8x3), which again makes no sense and should be replaced by a single experiment on the largest possible dataset.

Please explain why we need results from 4x3,8x3, etc and not simply report all results from the largest "slice", 16x3?

==

Section 3.2: We read that (IIUC) restoring *context* tokens reduces drift on downstream-layer *context* tokens more than restoring masked-object tokens. Why is that surprising? Surely it is to be expected that a given token attends mostly to its own location in the previous layer, and therefore that reinstating *context* tokens from previous layers would be more helpful (for context tokens of latter layers) than reinstating *object* tokens?

It would be the opposite result that would be surprising!

If my understanding of this passage is wrong please clarify it.

==

In multiple passages, the authors use words like "win" and "advantage" to describe what seems to be larger *damage* (e.g. 3.2, "they win on most images", where "win" apparently means more drift). This is exctremely confusing since the paper is suposedly about *repair*, so one looks in vain for some evidence of increased repair. Please rephrase to state explicitly "cause stronger drift" or something like that.

==

3.2, paragraph 2: IIUC, the authors claim that masking the *object* causes more drift in the *context* downstream tokens than masking the context itself. Is this a correct understanding? If so, it should be stated more explicitly.

As far as I can tell this is the one genuinely non-obvious result in the paper.

==

Section 3.3 is particularly aggravating.

We are given the following remarkably detailed claim: "A full-depth causal sweep with zero-overlap controls reveals three regimes: an early
local token-handling phase, a mid-depth rise, and a sharply peaked late repair window centered on layers 28–30."

As far as I can tell, the actual evidence supporting this claim is that reinstating more values, and reintating them from later layers, causes more repair (more reduction in drift) in the downstream layers! Which is exactly what one would expect?

I am particularly struck by the inclusion of the "three-site group" that includes resintating (some) values for layers 29, 30 and 31. But the "late context rescue" is precisely measured on layers 30 and 31 ! So reinstating values in some layers reduces drift in the same layers, more so that reinstating in other layers?

This is not "evidence" for "phases", "rises" or a "late repair window". Unless the description is mangled, this is an absolutely obvious finding.

==

Figure 5 should come up much earlier in the paper, perhaps as Figure 1 or 2, since it represents the only non-trivial finding in the paper (stronger impact of masking object than context on late context tokens). Also it finally explains what the various masks look like.

==

Various experiments confirm the previous findings. They seem fine on a cursory glance.

==

The Discussion summarizes the various strong claims made in the paper, which again are mostly unsupported by the utterly expected findings.

Authors claim to observe different behavior in early, moddle and late layer, and to find a "bottleneck" at layer 29 - there is no evidence anywhere in the paper to suport any such a "bottleneck". All that was reported in the paper was that reinstating later tokens, and more of them, reduced drift in later tokens (sometimes in the very same layer!) more.

Unless I missed it, the Discussion seems to contain no mention of the "semantic advantage" finding (stronger impact of masking object than context on late context tokens), which again (if confirmed) would be the one significant, non-trivial finding in the paper. Instead it is entirely devoted to this "semantic repair" concept which seems to bear no relation to the actual reported results.

==

The final sentence of the Discussion summarizes the problem with the paper:

"Our strongest conclusion is that JEPA implements a specific late repair circuit for handling semantic uncertainty, and that this circuit is consistent with a world-model view where the encoder prepares a context from which the predictor infers the hidden region’s latent."

This is basically fantasy, since nothing in the paper supports this claim.

**Requested Changes:**

Given the detailed review provided above, and unless the authors can somehow reframe their entire argument in a way that makes sense, it should be clear that no corrections would make the paper acceptable.

Instead, a qualified human should carefully go over the claims and the evidence, observe the enormous gap between them, and rewrite the entire paper.

---

### Note · Authors · 2026-04-27

I have read and agree with the venue's withdrawal policy on behalf of myself and my co-authors.